# Internode Length Is Correlated with GA_3_ Content and Is Crucial to the Harvesting Performance of Tea-Picking Machines

**DOI:** 10.3390/plants12132508

**Published:** 2023-06-30

**Authors:** Yao Luo, Qianqian Yu, Yinghua Xie, Chaojie Xu, Letian Cheng, Qing Shi, Yeyun Li, Xianchen Zhang, Zhougao Shen

**Affiliations:** State Key Laboratory of Tea Plant Biology and Utilization, Anhui Agricultural University, Hefei 230036, China; luoyao@stu.ahau.edu.cn (Y.L.); yuqianqian@stu.again.edu.cn (Q.Y.); 21720068@stu.ahau.edu.cn (Y.X.); xuchaojie@pinyuerp.com (C.X.); 106210431@stu.ahau.edu.cn (L.C.); 13170277328@163.com (Q.S.); liyeyun360@163.com (Y.L.)

**Keywords:** tea, machine picking, GAs, internode length

## Abstract

High labor costs and labor shortages are limiting factors affecting the tea industry in Anhui Province. Thus, exploiting the full mechanization of shoot harvesting is an urgent task in the tea industry. Tea quality is greatly influenced by the integrity rate of tea leaves; therefore, it is important to choose tea cultivars suitable for machine picking. In this study, seven tea cultivars were used to investigate the relationship between internode length and blade angle with respect to newly formed tea shoots and machine harvesting in field experiments (Xuanchen City, Kuiling village) conducted throughout the year (in the autumn of 2021, in the early spring of 2022, and in the summer of 2022). Our results showed that the internode length (L2 or L4) had a significant and positive correlation with the integrity rate of tea buds and leaves in seven tea cultivars over three seasons. However, no significant correlation was found between the blade angle and the integrity rate of tea buds and leaves. In addition, a strong and positive correlation was found between the levels of GA_1_ (R2 > 0.7), GA_3_ (R2 > 0.85), and IAA (R2 > 0.6) regarding the internodes and internode lengths of the seven tea cultivars. Moreover, the relative expression levels of *CsGA20ox*, *CsGA3ox1*, and *CsGA3ox2* in Echa1 (the longer internode) were significantly higher compared with those in Zhenong113 (the shorter internode). Overall, our results show that the internode length is an important factor for the machine harvesting of tea leaves and that the level of GA3 is strongly associated with internode length.

## 1. Introduction

The tea plant (*Camellia sinensis* (L.) O. Kuntze)) is a perennial woody economic crop [1]. In the main tea-growing district, tea seasons are commonly divided into spring (late March and April), summer (May, June, and July), and autumn (August and September) [2]. Tender young shoots consisting of a bud with one expanding leaf or two leaves are harvested in the early spring to produce the highest-quality tea. Tea products are the second most popular non-alcoholic beverages worldwide due to their refreshing tastes, characteristic aromas, and potential health benefits [3]. Traditionally, tea harvesting has been performed by hand [4]. Hand plucking reduces the occurrence of broken leaves and ensures that the most economically significant parts are harvested [5]. However, the manual harvesting of fresh tea leaves is an important task, and it is also the most labor-intensive and time-consuming process in the production and management of tea gardens [6]. With the increasing labor cost and production quality requirements demanded by tea producers, the full mechanization of shoot harvesting is an inevitable trend for the sustainable development of the tea industry [7].

In recent years, tea-plucking machines have been applied in the automatic picking of tea sprouts at home and abroad; these machines have been most widely used to improve picking efficiency and reduce picking costs [8]. Lin et al. [9] developed a system that guides operators of ridable tea-plucking machines in real time and collects images of tea canopies for tea plantation management. Yan et al. [10] used a lightweight convolutional neural network dubbed MC-DM (Multi-Class DeepLabV3+ MobileNetV2 (Mobile Networks Vision 2)) to improve the accuracy of picking point identification. A recent study reported that a robotic harvesting system incorporating the YOLOv3 network could quickly and accurately identify tea shoots, thus increasing harvesting success rates and shortening average plucking times [11].

As mentioned above, the development of mechanized picking equipment has significantly improved harvesting efficiency; however, such equipment currently suffers from many shortcomings. Firstly, due to the focus on small targets such as tea buds, there is great difficulty in the extraction of picking points, which will degrade the integrity of the sprouts and reduce the economic value of premium tea [5]. In addition, tea shoots grow densely at different heights in different regions; hence, machine picking easily damages the bodies of tea plants and reduces the tea yield in the upcoming year [12]. To ensure the integrity of bud identification, it is important to propose a potential standard for picking tea buds at harvesting time. This was the first aim of our study.

Internode distance is an important standard determining the suitability of tea plants for mechanical picking [13]. Growth-promoting phytohormones play a pivotal role in cell growth and internode elongation, thus regulating plant height [14]. GA, as a key modulator of internode elongation, mediates cellular processes and internode elongation in sugarcane stem growth [15]. When comparing the dwarf cultivar *Cenchrus purpureus* cv. Mott (CpMott) and the tall cultivar *C. purpureus* cv. GuiMinYin (CpGMY), it was observed that the decreased expression levels of genes involved in gibberellin biosynthesis resulted in a greater decrease in the accumulation of active GA_1_ in CpMott [16]. A recent study [17] reported that the exogenous application of GA_3_ can restore the dwarf phenotype of the GA-deficient mutant Oshls1-3, indicating that GA_3_ regulates rice height. In addition, gibberellin and auxin may lead to the differentiation of cell growth in the internode cell division zone and the cell elongation zone of bamboo [18]. However, whether gibberellin or auxin is strongly associated with tea internode length is insufficiently understood. Thus, the second aim of this study was to analyze the possible influence of phytohormones on internode length of tea plants.

As mentioned above, exploiting the full mechanization of tea shoot harvesting is an urgent task in the tea industry. A previous study reported the selection of a suitable machine-harvestable tea cultivar in Hangzhou [19]. In addition, the effect of the internode length and blade angle of newly formed tea shoots on the integrity of the shoots after being mechanically picked was investigated [20]. However, limited cultivar numbers, time, and experiment area are detrimental to the selection of machine-harvesting standards and tea cultivars sufficient for meeting the goal of “machine replacement”. In this study, experimental harvesting was conducted in a field in Xuanchen City, Kuiling Village, throughout the year (in the autumn of 2021, in the early spring of 2022, and in the summer of 2022) to evaluate the standards for the mechanization of tea plucking with regard to seven tea cultivars. Our results show that internode length is an important factor related to the integrity rate of buds and leaves after being harvested by a picking machine. Additionally, the influence of phytohormones on internode length was analyzed.

## 2. Results

### 2.1. New Shoot Growth Analysis

The seven tea cultivars grown in a natural environment are shown in Figure 1.

In addition, different harvest standards are shown in Figure 2.

As shown in Table 1, Table 2 and Table 3, dwarf growth was observed for Zhenong113, along with the shortest new shoot length and length of the internode. The greatest new shoot and internode lengths observed were those of Zhengcha111 in all three analyzed seasons. However, no similar trends were found regarding blade angle for the seven cultivars. For example, the blade angle of Zhongcha111, with the greatest new shoot length and internode length, was significantly lower than that of Zhenong113.

### 2.2. The Effect of Internode Length and Blade Angle on Machine-Picking Efficiency

To establish a scientific evaluation system for the machine-harvested tea cultivars, the effects of the machine harvesting of tea on the integrity rate of the tea buds and leaves were analyzed. Therefore, the samples were divided into groups consisting of a bud, one bud and one leaf, one bud and two leaves, one bud and three leaves, one bud and four leaves, and others (Figure 3).

As shown in Table 4, the highest rate of broken buds and leaves was observed for Zhenong113 in the autumn, while the highest rates of intact leaf buds were observed for Zhongcha111 and Echa1, which were significantly higher, i.e., 90% and 65.80%, than the same rate for Zhenong113.

Similar results showed that the highest rates of intact leaf buds in the spring were observed for Zhongcha111 and Echa1, while this value for Zhenong113 was only 56.74% (Table 5). In addition, the rate of intact leaf buds for most of the tea cultivars in the summer increased to above 80%; however, the lowest integrity rate of buds and leaves was still presented by Zhenong113 (Table 6).

To further clarify the influence of internode length or blade angle on machine-harvested tea plants, the correlation between internode length and blade angle and the integrity rate of buds and leaves was analyzed. As shown in Figure 4A, in the autumn, a significant positive relationship between internode length (L1, L2, and L4) and the integrity rate of buds and leaves was observed. Additionally, a significant correlation was found between L2 and the integrity rate of buds and leaves in the spring (Figure 4B) and summer (Figure 4C). However, the correlation coefficient between the blade angle and the integrity rate of buds and leaves in the three seasons indicated that there was no significant correlation.

### 2.3. GAs and IAA Content along Internode Length

As mentioned above, GAs or IAA may be key factors in stem elongation, and a significant correlation was found between L2 and the integrity rate of buds and leaves through the year. Thus, the levels of GAs and IAA of the internodes (L2) in the spring were detected. As shown in Table 7, the longer internodes (L2) in Echa1 and Zhongcha111 exhibited higher levels of GA_1_, GA_3_, and IAA; additionally, lower accumulation levels of GA_1_, GA_3_, and IAA were detected in Zhenong113, which has a shorter internode length (L2). However, a similar tread was not found regarding GA_4_.

To further clarify the role of hormones in determining internode length, the correlation between hormone levels and internode length (L2) was analyzed. A higher correlation was observed between the levels of GA_1_ (Figure 5A), GA_3_ (Figure 5B), and IAA (Figure 5D) and internode length (L2) in the spring. The highest correlation observed was between GA_3_ and internode length (R^2^ > 0.85). However, in a manner consistent with the GA_4_ content among the seven cultivars, there was no significant correlation between GA_4_ level and internode length (L2) (Figure 5C).

Moreover, the expression levels of *CsGA20ox*, *CsGA3ox1*, and *CsGA3ox2* in GAs signaling pathways were analyzed in two contrasting cultivars (namely, long-internode and short-internode cultivars). The results showed that the expression of *CsGA20ox*, *CsGA3ox1*, and *CsGA3ox2* in Echa1 was significantly higher (by 189%, 384%, and 186%, respectively) than that in Zhenong113 (Figure 6).

## 3. Discussion

Traditional high-quality tea plucking by hand is time consuming, labor-intensive, and costly. High-quality tea harvesting must be completed within a month in the early spring, so a considerable amount of work is required within a short time frame [21]. Moreover, unlike other crops (e.g., apples or strawberries), tea shoots grow densely at different heights, thus necessitating the consideration of their path of motion during harvesting. Therefore, tea-harvesting robots must be highly efficient.

### 3.1. A Scientific and Reasonable Comprehensive Evaluation System of Machine-Harvested Tea

Tea quality is strongly associated with the integrity rate of tea buds and leaves. An increasing amount of work concerning the screening of tea cultivars to determine their suitability for machine harvesting is being conducted. For example, the length of one bud and one/two leaves, the growth angle of the second/third leaf, the number of leaf buds, etc., were determined to screen for core evaluation indicators [22]. The presently discussed field experiment was conducted to characterize the relationship between different harvesting standards and the integrity rate of tea buds and leaves, i.e., buds with one, two, or three young expanding leaves and with high yields and growth [23]. However, the use of a limited number of cultivars or a short timeframe is inadequate for the selection of machine-harvesting standards and tea cultivars that allow the goal of “machine replacement” to be met. In our study, the results showed that internode length (Zhongcha111 and Echa1) was closely related to a higher integrity rate of the tea buds and leaves in the seven tea cultivars throughout the studied years. A similar result showed that Chuancha2, Zhongcha108, and Mabianlv1, with greater internode lengths, exhibited higher integrity rates for their tea buds and leaves [24]. Therefore, we inferred that internode length may be applied to evaluate the adaptability of machine-harvested tea cultivars.

In addition, the relationship between the blade angle of newly formed tea shoots (β1 or β2) and machine picking was analyzed. In the spring of 2022, the β1 and β2 of most of the tea cultivars with different internode lengths were near 40 °C, exhibiting no significant differences (Table 1); thus, the correlation coefficient was less than 0.2 (Figure 4). Similar results were obtained in the autumn of 2021 and the summer of 2022. Therefore, we inferred that blade angle may not be an important parameter for machine picking. To draw further conclusions, more tea cultivars were employed to analyze the possible relationship in future studies.

### 3.2. The Influence of Hormones on Internode Distance

As mentioned above, the internode length plays an important role in machine picking. The activity of GAs is believed to directly influence plant elongation through cell growth regulation [25]. A previous study [26] reported that a 13-hydroxylation pathway promoted an increase in GA_1_ levels, which sustained poplar shoot apex development. Similar results were reported regarding the role of GA_3_ or GA_4_ in potato (Solanum tuberosum) [27] or yam (Dioscorea opposita) [28] growth.

The major bioactive GAs are GA_1_, GA_3_, GA_4_, and GA_7_, while the others are the precursors or deactivated forms of bioactive GAs [29]. Therefore, GA_1_, GA_3_, GA_4_, and GA_7_ (undetected) were examined in the internodes of seven tea cultivars in the spring of 2022. Our results showed that Fuzao2, Echa1, and Zhongcha111 with longer internodes (L2) presented higher levels of GA_1_ and GA_3_, but opposite results were found regarding Fuyun6 and Zhenong113 with shorter internode lengths (L2). In addition, a highly significant correlation (R^2^ > 0.85) was observed between the levels of GA_3_ (Figure 5B) and internode length (L2) in the spring. Shan et al. (2021) reported that the GA_3_ level in stems is an important factor affecting soybean internode elongation [30]. Similar results showed that the exogenous application of GA_3_ enhanced the lignification of the xylem cell wall in turnip (*Brassica rapa* var. *rapa*) [31]. In accordance with the previous study, we inferred that GA_3_ levels may be an important component associated with internode length.

To further clarify the role of GAs in tea plant internode length, certain gene expression levels in the GA pathways were analyzed. Our previous study reported that the *CsGA20ox* and *CsGA3ox* genes may be closely related to the bioactive GA levels in tea [32]. Therefore, in our study, the expression levels of *CsGA20ox*, *CsGA3ox1*, and *CsGA3ox2* were analyzed. Our results showed higher expression levels of *CsGA20ox*, *CsGA3ox1*, and *CsGA3ox2* in Echa1, which has longer internodes compared with those of Zhenong113 (Figure 6). Similarly, another study showed that a GA-mediated signaling pathway regulated rice stem elongation by activating *OsGA2ox3* expression [33]. Paciorek et al. [34] reported that the inhibition of *ZmGA20ox3* and *ZmGA20ox5* resulted in a reduction in GA levels, leading to the production of a short-statured maize ideotype. On the contrary, the overexpression of *ZmGA20ox1* or *PsGA3ox1* increased the length of maize stems [35] and pea internodes [36], respectively. The above results further indicate that *CsGA20ox*, *CsGA3ox1*, and *CsGA3ox2* may play important roles in tea internode growth. However, some molecular mechanisms relating to the roles of *CsGA20ox*, *CsGA3ox1*, or *CsGA3ox2* in tea internode elongation are scarcely understood, and we will carry out more work in this regard in future studies.

In addition, auxin (IAA) regulates stalk development through complex signal transduction pathways [37]. In our study, a significant correlation between internode lengths and IAA levels was detected, which implied that IAA may play an important role in internode growth. Kou et al. (2021) also found that IAA induced an enhancement in Chinese cabbage flower stalk elongation, while uniconazole (a GA synthesis inhibitor) and N-1-naphthylphthalamic acid (NPA) (an auxin transport inhibitor) significantly impaired stalk expansion [38].

Moreover, to further explore the influence of GAs and IAA on internode length, the ratio of GAs to IAA was analyzed. The ratio of GAs to IAA in the tea cultivar Zhongcha111, with a longer internode length, was about 20.24%, while these ratios were only 13.79 % and 11.7 % in Longjinchangye and Zhenong113, which have shorter internode lengths (Table 2). These results further indicate that GAs may play a more important role in internode length when compared with IAA.

## 4. Materials and Methods

### 4.1. Design of the Experiment

Seven cultivars of tea plants (Zhongcha111, Echa1, Fuzao2, Yingshuang, Longjingchangye, Fuyun6, and Zhenong113) were employed in the current study. All tea plants (70–75 cm in height; seven years old) were grown in Hongling County (longi-tude: 118°76′25′′ and latitude: 30°95′21′′), Xuanchen City, Anhui Province.

A randomized complete block design incorporating 7 tea cultivars was implemented in the field experiments. Each treatment contained 3 rows of tea plants with 35 tea plants per row, and the area was approximately 40 m^2^. A pre-experiment involving pruning management was carried out on 2 March 2021. The field experiments were carried out throughout the year (in the autumn of 2021, in the early spring of 2022, and in the summer of 2022). During the experimental period, the tea plants were irrigated, and weeds were mechanically removed during the growing season. Fertilizer was applied at a rate of 1500 kg ha^−1^ cake fertilizer in the autumn.

The tea plants in all treatments were subjected to light pruning, and were generally kept trimmed to a convenient picking height of 70 to 75 cm, on 24 August in 2021 and on 16 February and 23 July in 2022. A total of 450 kg ha^−1^ urea sample was applied after 5 days of pruning. When the tea plants sprouted new buds, the tea leaves were plucked using a one-person electric portable handheld tea-plucking machine on 20–25 September in 2021 and on 8–16 April and 28 August–2 September in 2022.

### 4.2. Harvest Standards

There were two harvest standards applied in this study, as shown in Figure 2, namely, internode length (referred to as L1, L2, L3, and L4) and blade angle (β1 and β2), which were applied in the spring and summer or autumn, respectively. The harvest standards (treatment) were determined visually according to the number of expanding leaves. First, 50 new branches were harvested by hand, and each harvest standard was measured. Then, canopies with an area of 1 m^2^ were plucked using a one-person electric portable handheld tea-plucking machine, and those with one bud and one leaf, one bud two leaves, one bud and three leaves, one bud and four leaves, and other parts were classified and counted. In addition, standard leaf buds and intact leaf buds were counted using the following formulae:①Standard leaf bud = one bud and two leaves + one bud and three leaves;②Intact leaf buds = one bud and one leaf + one bud and two leaves + one bud and three leaves + one bud and four leaves.

### 4.3. Gibberellin and IAA Extraction and Quantification

Briefly, 1.0 g frozen samples were ground with 20 mL of acetonitrile solution containing 1% formic acid for 2 min. The supernatant was then collected after centrifugation at 1000× *g* for 5 min at 4 °C. About 4 mL of supernatant was then transferred to an Eppendorf tube containing 300 mg of anhydrous magnesium sulfate and 100 mg of C18 and then centrifuged at 10,000× *g* for 5 min at 4 °C. The supernatant was diluted to 1mL using methyl alcohol. The analysis of the endogenous hormones was performed using a liquid chromatography–series mass spectrometer, as described by Jing et al. [39] and Wang et al. [40].

### 4.4. RNA Extraction and Quantitative Real-Time PCR

qRT-PCR was performed by following a three-step method [1]. Briefly, samples from the tea internodes were collected to isolate their total RNA. cDNA (complemen-tary cDNA) was synthesized using the TAKARA PrimeScript II 1st Strand cDNA Synthesis Kit. The detection conditions for *CsGA20ox* (TEA011661), *CsGA3ox1* (TEA012353), and *CsGA3ox2* (TEA001361) were 95 °C for 30 s; 40 cycles of 95 °C for 5 s, followed by 60 °C for 30 s; and 72 °C for 30 s, respectively. *CsGADPH* was used as a reference for qPCR data analysis. All of the primers used in this study are shown in Appendix A.

### 4.5. Statistical Analysis

All the measurements were based on different replicate samples. All the data analyses were performed using SPSS 26.0 software. Data are presented as the mean ± SD. The statistical analysis included a one-way analysis of variance (ANOVA), and significant differences between the means were tested using Duncan’s multiple range test at 95% confidence.

## 5. Conclusions

Seven tea cultivars were used to investigate the relationship between the internode length or blade angles of newly formed tea shoots and the integrity of buds and leaves in field experiments conducted over one year. Longer internode length was strongly associated with the integrity of buds and leaves, and the level of GA_3_ was strongly related to internode length. Our study indicated that the internode length is crucial to the harvesting performance of tea-picking machines. Therefore, in future studies, tea cultivars with longer internodes will need to be cultivated to determine which cultivars are suitable for machine picking.

## Figures and Tables

**Figure 1 plants-12-02508-f001:**
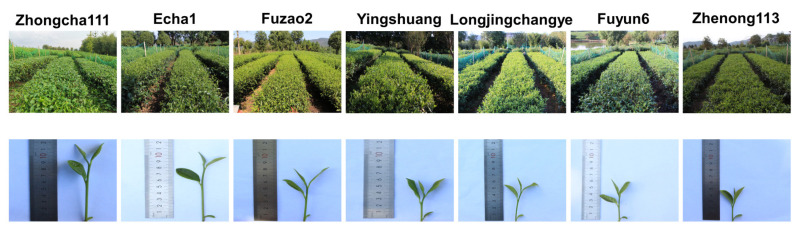
Seven tea plant cultivars analyzed in the field experiments.

**Figure 2 plants-12-02508-f002:**
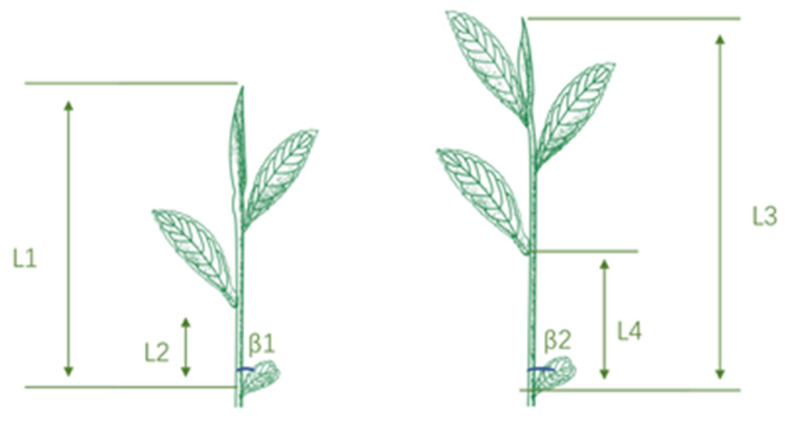
The internode lengths and blade angles of tea plants. Note: **L1**, **L2**, **L3** and **L4** were internode lengths; **β1** and **β2** indicated blade angles.

**Figure 3 plants-12-02508-f003:**
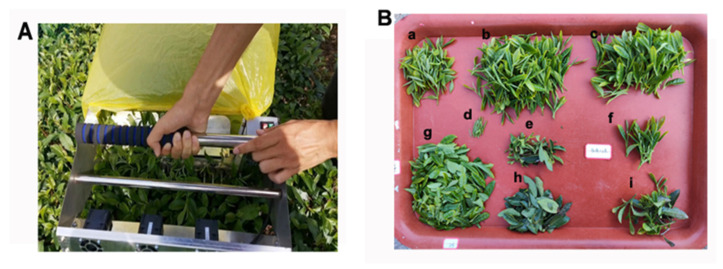
The parts of tea in the samples. Machine picking and the different parts of the samples shown in (**A**,**B**), respectively. Notes: **a**–**i** indicate the following: one bud and one leaf, one bud and two leaves, one bud and three leaves, one bud, old stem, non-standard bud leaves, damaged new leaves, damaged old leaves, and others, respectively.

**Figure 4 plants-12-02508-f004:**
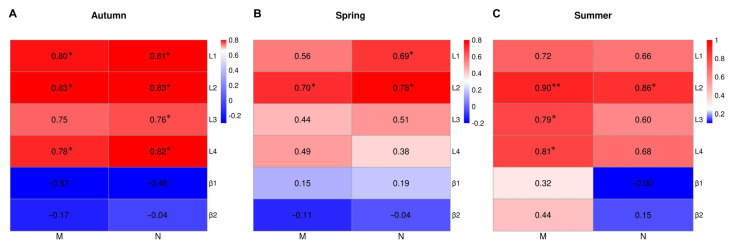
The relationship between bud and leaf integrity following machine picking and internode length or blade angle for the seven tea cultivars. Notes: M and N indicate the standard rate of buds and leaves and the intact rate of buds and leaves, and the data in the heat map indicate the correlation coefficient between M or N and the internode length or blade angle of the seven tea cultivars in 2021 autumn (**A**), 2022 spring (**B**) and 2022 summer (**C**). β1 and β2 indicated that blade angle and * indicated that significant differences at the level of *p* < 0.05.

**Figure 5 plants-12-02508-f005:**
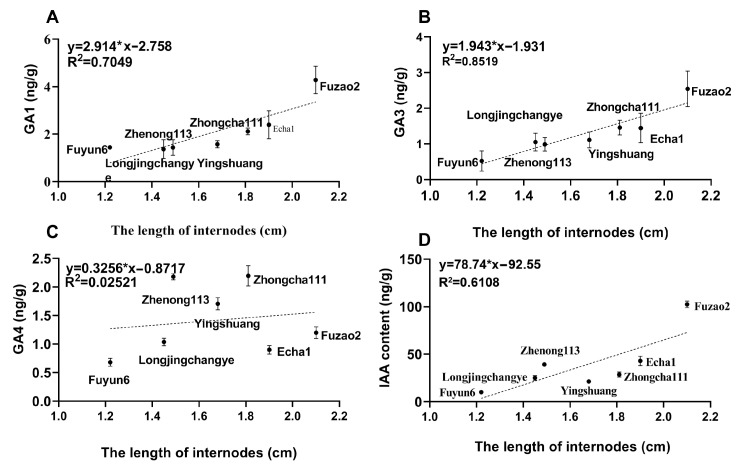
The relationship between internode length (L2) and hormones in internodes.

**Figure 6 plants-12-02508-f006:**
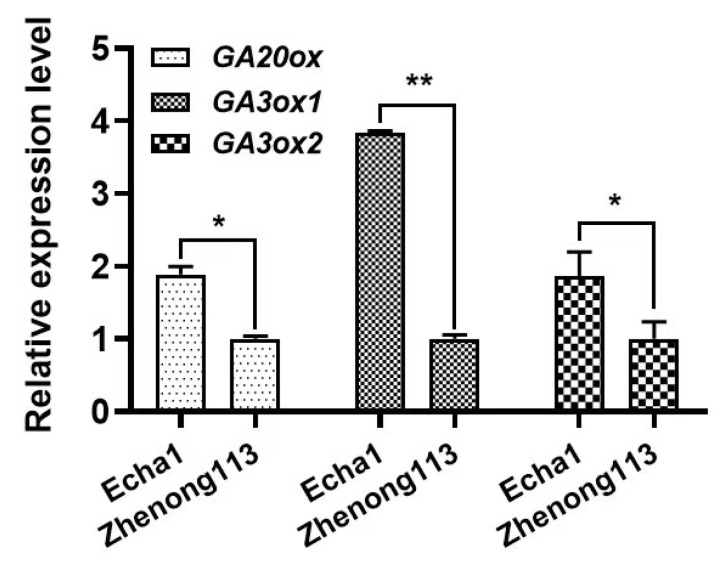
The expression levels of CsGA20ox, CsGA3ox1, and CsGA3ox2 in the two cultivars. Note: *, ** indicated that significant differences at the level of *p* < 0.05.

**Table 1 plants-12-02508-t001:** Morphological characteristics of newly formed tea shoots of different tea cultivars in the autumn of 2021.

Cultivar	One Bud and Two Leaves			One Bud and Three Leaves		
	L1 (cm)	L2 (cm)	β1	L3 (cm)	L4 (cm)	β2
Zhongcha111	9.22 ± 0.03 ^a^	4.16 ± 0.04 ^a^	22.86 ± 0.29 ^e^	12.2 ± 0.06 ^a^	4.37 ± 0.08 ^a^	24.61 ± 0.23 ^e^
ECha1	7.9 ± 0.01 ^b^	3.6 ± 0.08 ^b^	33.39 ± 0.3 ^a^	10.46 ± 0.45 ^c^	3.92 ± 0.05 ^b^	39.43 ± 0.34 ^a^
Fuzao2	8.02 ± 0.07 ^b^	3.33 ± 0.04 ^c^	33.27 ± 0.45 ^a^	10.89 ± 0.09 ^b^	3.99 ± 0.05 ^b^	33.28 ± 0.33 ^b^
Yingshuang	6.43 ± 0.09 ^c^	2.95 ± 0.12 ^d^	23.05 ± 0.12 ^e^	8.01 ± 0.09 ^d^	3.26 ± 0.01 ^c^	25.81 ± 0.17 ^d^
Longjingchangye	5.94 ± 0.05 ^d^	2.67 ± 0.09 ^e^	25.14 ± 0.35 ^d^	7.93 ± 0.04 ^d^	2.61 ± 0.04 ^d^	24.84 ± 0.41 ^e^
Fuyun6	5.66 ± 0.21 ^e^	2.05 ± 0.09 ^f^	29.8 ± 0.48 ^c^	7.86 ± 0.1 ^d^	2.51 ± 0.04 ^e^	30.38 ± 1.13 ^c^
Zhenong113	4.89 ± 0.02 ^f^	1.67 ± 0.03 ^g^	32.55 ± 0.65 ^b^	6.83 ± 0.05 ^e^	2.02 ± 0.09 ^f^	29.54 ± 0.33 ^c^

Note: Different lowercase letters indicate significant differences at the level of *p* < 0.05.

**Table 2 plants-12-02508-t002:** Morphological characteristics of newly formed tea shoots of different tea cultivars in the spring of 2022.

Cultivar	One Bud and Two Leaves			One Bud and Three Leaves		
	L1 (cm)	L2 (cm)	β1	L3 (cm)	L4 (cm)	β2
Zhongcha111	6.87 ± 0.39 ^a^	1.81 ± 0.09 ^b^	39.29 ± 3.58 ^ab^	9.1 ± 0.34 ^ab^	1.93 ± 0.21 ^b^	41.04 ± 1.66 ^ab^
ECha1	6.89 ± 0.61 ^a^	1.9 ± 0.08 ^ab^	42.64 ± 0.68 ^a^	10.05 ± 0.86 ^a^	1.99 ± 0.09 ^b^	41.1 ± 0.34 ^ab^
Fuzao2	6.68 ± 0.3 ^a^	2.1 ± 0.22 ^a^	34.01 ± 2.4 ^b^	7.84 ± 0.6 ^bcd^	2.01 ± 0.33 ^b^	37.05 ± 2.13 ^b^
Yingshuang	5.46 ± 0.22 ^bc^	1.68 ± 0.04 ^bc^	41.7 ± 5.44 ^a^	7.97 ± 1.04 ^bcd^	3.32 ± 0.44 ^a^	25.62 ± 7.76 ^c^
Longjingchangye	4.95 ± 0.25 ^c^	1.45 ± 0.1 ^cd^	35.47 ± 4.32 ^ab^	6.59 ± 0.86 ^d^	1.3 ± 0.11 ^c^	39.78 ± 3.72 ^ab^
Fuyun6	5.71 ± 0.35 ^b^	1.22 ± 0.19 ^d^	42.68 ± 4.94 ^a^	8.5 ± 0.94 ^bc^	2.12 ± 0.42 ^b^	45.58 ± 2.27 ^a^
Zhenong113	5.78 ± 0.49 ^b^	1.49 ± 0.2 ^c^	38.34 ± 2.88 ^ab^	7.46 ± 0.21 ^cd^	1.26 ± 0.22 ^c^	42.93 ± 1.44 ^ab^

Note: Different lowercase letters indicate significant differences at the level of *p* < 0.05.

**Table 3 plants-12-02508-t003:** Morphological characteristics of newly formed tea shoots of different tea cultivars in the summer of 2022.

Cultivar	One Bud and Two Leaves			One Bud and Three Leaves		
	L1 (cm)	L2 (cm)	β1	L3 (cm)	L4 (cm)	β2
Zhongcha111	5.83 ± 0.19 ^a^	2.5 ± 0.16 ^a^	30.85 ± 3.23 ^a^	7.76 ± 0.84 ^a^	2.67 ± 0.58 ^a^	35.29 ± 0.68 ^bc^
ECha1	5.5 ± 0.13 ^ab^	2.18 ± 0.09 ^a^	33.22 ± 2.15 ^a^	8.04 ± 0.6 ^a^	2.51 ± 0.21 ^a^	39.05 ± 3.89 ^ab^
Fuzao2	5.49 ± 0.06 ^ab^	1.32 ± 0.21 ^bc^	30.77 ± 1.05 ^a^	6.85 ± 0.5 ^abc^	1.41 ± 0.23 ^b^	31.31 ± 3.71 ^cd^
Yingshuang	4.69 ± 0.56 ^cd^	1.65 ± 0.28 ^b^	34.87 ± 0.93 ^a^	7.29 ± 1.27 ^ab^	1.86 ± 0.31 ^b^	35.1 ± 2.22 ^bc^
Longjingchangye	4.32 ± 0.24 ^d^	1.58 ± 0.17 ^b^	26.13 ± 3.23 ^b^	5.27 ± 0.64 ^d^	1.3 ± 0.12 ^b^	27.26 ± 1.48 ^d^
Fuyun6	5.06 ± 0.31 ^bc^	1.64 ± 0.21 ^b^	35.04 ± 2.53 ^a^	6.04 ± 0.64 ^bcd^	1.32 ± 0.35 ^b^	40.37 ± 2.83 ^a^
Zhenong113	3.48 ± 0.31 ^e^	1.18 ± 0.08 ^c^	32.42 ± 3.14 ^a^	5.82 ± 0.39 ^cd^	1.49 ± 0.09 ^b^	35.63 ± 0.58 ^bc^

Note: Different lowercase letters indicate significant differences at the level of *p* < 0.05.

**Table 4 plants-12-02508-t004:** The composition of different tea cultivars for which fresh leaves were picked in the autumn of 2021.

Cultivar	Bud and Leaves				Machine-Harvest Waste			Standard Bud Leaf (%)	Intact Bud and Leaves (%)
	One Bud and One Leaf (%)	One Bud and Two Leaves (%)	One Bud and Three Leaves (%)	One Bud and Four Leaves (%)	Other (%)	Broken Leaves (%)	Branch (%)		
Zhongcha111	0.44 ± 0.17 ^de^	43.45 ± 5.9 ^a^	34.46 ± 11.81 ^a^	2.5 ± 2.54 ^c^	4.68 ± 4.94 ^c^	14.48 ± 4.17 ^b^	0 ± 0	77.9 ± 6.16 ^a^	80.84 ± 8.42 ^a^
ECha1	2.53 ± 0 ^b^	33.01 ± 1.29 ^b^	29.39 ± 6.33 ^a^	5.59 ± 1.59 ^bc^	15.84 ± 5.32 ^b^	13.63 ± 2.89 ^b^	0 ± 0	62.4 ± 6.22 ^b^	70.53 ± 7.79 ^ab^
Fuzao2	0.06 ± 0.05 ^e^	20.71 ± 2.01 ^d^	34.18 ± 5.86 ^a^	12.38 ± 1.09 ^a^	15.67 ± 1.82 ^b^	14.75 ± 2.05 ^b^	2.24 ± 3.87	54.9 ± 7.86 ^b^	67.34 ± 6.77 ^b^
Yingshuang	1.35 ± 0.11 ^c^	26.1 ± 2.14 ^cd^	35.76 ± 5.26 ^a^	6.66 ± 2.61 ^b^	8.68 ± 5.13 ^bc^	21.45 ± 4.37 ^a^	0 ± 0	61.86 ± 7.08 ^b^	69.87 ± 9.1 ^ab^
Longjingchangye	1.43 ± 0.64 ^c^	22.15 ± 2.44 ^d^	32.47 ± 3.29 ^a^	5.45 ± 0.8 ^bc^	12.84 ± 4.22 ^bc^	25.66 ± 0.44 ^a^	0 ± 0	54.62 ± 5.61 ^b^	61.5 ± 4.62 ^b^
Fuyun6	6.52 ± 0 ^a^	30.97 ± 3.49 ^bc^	27.5 ± 2.86 ^a^	2.64 ± 2.3 ^c^	6.22 ± 5.53 ^bc^	25.22 ± 0.98 ^a^	0.93 ± 1.61	58.46 ± 4.94 ^b^	67.62 ± 4.91 ^b^
Zhenong113	0.73 ± 0.01 ^d^	24.45 ± 1.62 ^d^	14.34 ± 2.61 ^b^	3.03 ± 2.21 ^bc^	31.77 ± 7.15 ^a^	25.69 ± 3.67 ^a^	0 ± 0	38.79 ± 4.15 ^c^	42.54 ± 4.39 ^c^

Note: Different lowercase letters indicate significant differences at the level of *p* < 0.05.

**Table 5 plants-12-02508-t005:** The composition of different tea cultivars for which fresh leaves were picked in the autumn of 2021.

9	Bud and Leaves					Machine-Harvest Waste			Standard Bud Leaf (%)	Intact Bud and Leaves (%)
	One bud (%)	One Bud and One Leaf (%)	One Bud and Two Leaves (%)	One Bud and Three Leaves (%)	One Bud and Four Leaves (%)	Other (%)	Broken Leaves (%)	Branch (%)		
Zhongcha111	1.21 ± 0.25 ^a^	9.36 ± 3.2 ^abc^	28.98 ± 3.19 ^bc^	27.97 ± 4.66 ^ab^	9.19 ± 2.68 ^a^	3.63 ± 2.32 ^b^	16.98 ± 2.09 ^d^	2.67 ± 1.52 ^bc^	56.96 ± 3.35 ^a^	76.71 ± 3.5 ^ab^
ECha1	0.66 ± 0.18 ^abc^	10.33 ± 3.29 ^ab^	32.9 ± 1.29 ^ab^	25.21 ± 6.92 ^abc^	3.35 ± 1.49 ^b^	6.81 ± 2.84 ^ab^	16.14 ± 1.68 ^d^	4.61 ± 0.31 ^ab^	58.11 ± 5.74 ^a^	72.45 ± 4.02 ^abc^
Fuzao2	0.48 ± 0.36 ^bc^	7.51 ± 1.77 ^abcd^	31.53 ± 2.6 ^abc^	32.26 ± 6.1 ^a^	7.81 ± 1.89 ^a^	2.3 ± 0.23 ^b^	16.84 ± 0.88 ^d^	1.26 ± 0.94 ^c^	63.79 ± 3.69 ^a^	79.6 ± 1.48 ^a^
Yingshuang	0.9 ± 0.08 ^abc^	10.46 ± 2.42 ^ab^	36.14 ± 2.16 ^a^	21.37 ± 6.63 ^bcd^	0.6 ± 1.05 ^b^	3.05 ± 1.1 ^b^	23.81 ± 3.69 ^bc^	3.66 ± 1.07 ^abc^	57.51 ± 4.96 ^a^	69.48 ± 3.63 ^bc^
Longjingchangye	0.32 ± 0.24 ^bc^	4.54 ± 1.44 ^d^	27.16 ± 2.78 ^c^	32.73 ± 4.36 ^a^	10.9 ± 3.42 ^a^	1.91 ± 0.91 ^b^	20.8 ± 1.73 ^cd^	1.65 ± 0.78 ^c^	59.89 ± 1.59 ^a^	75.64 ± 3.22 ^ab^
Fuyun6	0.64 ± 0.18 ^abc^	7.23 ± 0.44 ^a^	32.1 ± 4 ^abc^	24.84 ± 4.95 ^abc^	3.04 ± 1.77 ^b^	7.88 ± 2.76 ^ab^	19.51 ± 4.26 ^cd^	4.75 ± 0.38 ^ab^	56.95 ± 6.2 ^a^	67.86 ± 6.96 ^c^
Zhenong113	1 ± 0.78 ^ab^	11.04 ± 3.7 ^abcd^	29.54 ± 4.45 ^bc^	14.8 ± 3.69 ^d^	0.36 ± 0.63 ^b^	12.53 ± 6.67 ^a^	24.95 ± 5.63 ^bc^	5.78 ± 0.47 ^a^	44.33 ± 6.78 ^b^	56.74 ± 2.75 ^d^

Note: Different lowercase letters indicate significant differences at the level of p < 0.05.

**Table 6 plants-12-02508-t006:** The composition of different tea cultivars for which fresh leaves were picked in the summer of 2021.

Cultivar	Bud and Leaves					Machine-Harvest Waste			Standard Bud Leaf (%)	Intact Bud and Leaves (%)
	One Bud (%)	One Bud and One Leaf (%)	One Bud and Two Leaves (%)	One Bud and Three Leaves (%)	One Bud and Four Leaves (%)	Other (%)	Broken Leaves (%)	Branch (%)		
Zhongcha111	0 ± 0 ^b^	3.77 ± 1.76 ^c^	33.35 ± 8.44 ^ab^	36.97 ± 0.9 ^a^	6.66 ± 6.63 ^bc^	6.05 ± 4.55 ^a^	9.87 ± 1.02 ^e^	3.34 ± 1.71 ^a^	70.32 ± 8.55 ^a^	80.75 ± 2.6 ^ab^
ECha1	0 ± 0 ^b^	3.62 ± 0.25 ^c^	31.71 ± 3.08 ^ab^	38.72 ± 3.03 ^a^	8.07 ± 1.26 ^bc^	5.91 ± 3.4 ^a^	11.97 ± 1.93 ^de^	0 ± 0 ^b^	70.43 ± 0.39 ^a^	82.12 ± 1.52 ^a^
Fuzao2	0.43 ± 0.26 ^a^	12.24 ± 3.22 ^a^	36.87 ± 4.94 ^a^	19.32 ± 5.03 ^d^	4.88 ± 6.13 ^bc^	1.88 ± 0.66 ^ab^	24.38 ± 3.81 ^b^	0 ± 0 ^b^	56.19 ± 2.34 ^cd^	73.74 ± 3.16 ^c^
Yingshuang	0 ± 0 ^b^	6.03 ± 1.82 ^bc^	37.01 ± 1.85 ^a^	30 ± 1.97 ^bc^	8.99 ± 3.17 ^b^	1.96 ± 1.64 ^ab^	15.5 ± 2.64 ^cd^	0.51 ± 0.88 ^b^	67.01 ± 2.63 ^ab^	82.03 ± 1.78 ^a^
Longjingchangye	0.14 ± 0.24 ^b^	4.94 ± 0.89 ^bc^	26.19 ± 5.21 ^b^	31.82 ± 0.68 ^b^	17.62 ± 4.64 ^a^	0.76 ± 1.32 ^c^	18.45 ± 2.35 ^c^	0.09 ± 0.16 ^b^	58.01 ± 4.6 ^bcd^	80.7 ± 2.18 ^ab^
Fuyun6	0 ± 0 ^b^	8.26 ± 0.96 ^b^	37.53 ± 3.06 ^a^	25.24 ± 2.84 ^c^	3.92 ± 2.25 ^bc^	5.35 ± 2.87 ^ab^	19.71 ± 2.56 ^c^	0 ± 0 ^b^	62.77 ± 5.45 ^abc^	74.94 ± 5.41 ^bc^
Zhenong113	0 ± 0 ^b^	12.83 ± 2.89 ^a^	37.35 ± 3.9 ^a^	13.92 ± 3.41 ^e^	0.9 ± 0.94 ^c^	4.57 ± 1.4 ^ab^	29.05 ± 2.82 ^a^	1.39 ± 2.41 ^ab^	51.26 ± 7.3 ^d^	64.99 ± 5.15 ^d^

Note: Different lowercase letters indicate significant differences at the level of *p* < 0.05.

**Table 7 plants-12-02508-t007:** The GAs and IAA content of the internodes (L2) in the seven tea cultivars.

Cultivar	Content (ng/g)			
	GA_1_	GA_3_	GA_4_	IAA
Zhongcha111	2.118 ± 0.14 ^b^	1.458 ± 0.21 ^b^	2.195 ± 0.18 ^a^	28.513 ± 2.65 ^d^
Echa1	2.395 ± 0.59 ^b^	1.444 ± 0.41 ^b^	0.898 ± 0.08 ^e^	42.959 ± 4.96 ^b^
Fuzao2	4.283 ± 0.58 ^a^	2.544 ± 0.5 ^a^	1.199 ± 0.1 ^c^	102.382 ± 3.52 ^a^
Yingshuang	1.577 ± 0.15 ^c^	1.113 ± 0.22 ^bc^	1.705 ± 0.11 ^b^	21.323 ± 1.75 ^e^
Longjingchangye	1.373 ± 0.4 ^c^	1.05 ± 0.25 ^bc^	1.037 ± 0.07 ^d^	25.092 ± 2.54 ^d^
Fuyun6	1.448 ± 0.07 ^c^	0.52 ± 0.28 ^d^	0.681 ± 0.07 ^g^	10.014 ± 1.11 ^f^
Zhenong113	1.442 ± 0.32 ^c^	0.989 ± 0.19 ^c^	2.18 ± 0.06 ^a^	39.206 ± 1.59 ^c^

Note: Different lowercase letters indicate significant differences at the level of *p* < 0.05.

## Data Availability

Not Applicable.

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
