# Peer review of "Internode Length Is Correlated with GA3 Content and Is Crucial to the Harvesting Performance of Tea-Picking Machines"

_plants, 2023, doi:10.3390/plants12132508_

Round 1

Reviewer 1 Report

This paper reported the relationship between internode length and machine pick up performance of tea plant, and found that GA3 was strongly associated with internode length. The results will help for breeding for machine plucking tea plant cultivars. But there were many major concerns need to be addressed before further consideration.

1. The writing need to be improved. There were so many oral statements rather than scientific language. And the logic needs to be re-organized.

2. Integrity rate and efficient of machine plucking tea shoots are not only influenced by internode length, also influenced by buds sprouting uniformity. Moreover, the angle between the stem and the mature leaf is an important factor that influence the integrity rate of machine plucking tea shoots rather than β1 and β2. So this parameter needs to be analyzed. By the way, the number of the tea plant cultivars was limited, I think the data wasn't suitable for drawing the conclusion.

3. The ratio of GAs and IAA also influences the internode length, so this parameter needs to be analyzed. 

4. Discussion section needs to be re-written. 

5. 'variety' needs to be replaced by 'cultivar' all through the text.

There were so many oral statements rather than scientific language.  And some grammar and spelling errors need carefully checked.

Author Response

  1. The writing need to be improved. There were so many oral statements rather than scientific language. And the logic needs to be re-organized.

Response: Thank you for your valuable suggestion. The manuscript has been polished by the MDPI to improve the presentation of this manuscript. In addition, logic has been re-organized in discussion.

  1. Integrity rate and efficient of machine plucking tea shoots are not only influenced by internode length, also influenced by buds sprouting uniformity. Moreover, the angle between the stem and the mature leaf is an important factor that influence the integrity rate of machine plucking tea shoots rather than β1 and β2. So this parameter needs to be analyzed. By the way, the number of the tea plant cultivars was limited, I think the data wasn't suitable for drawing the conclusion.

Response: Thank you for your positive suggestion.

1 Buds sprouting uniformity is an important component associated with machine plucking. However, buds sprouting uniformity were not counted in our study. Therefore, in our future, the contribution of buds sprouting uniformity on machine plucking tea shoots was analyzed.

2 In our study, the relationship between β1 or β2 and machine plucking was analyzed in line and discussed it in the revised manuscript (Line 267-274).

3 In our study, 7 tea cultivar were used to investigate the relationship between the internode length or blade angle of newly-formed tea shoots and machine-harvest in the field experiments. The reviewer think that the number of the tea plant cultivars was limited, and the data wasn't suitable for drawing the conclusion. We partly agreed with your positive suggestion. In 7 tea cultivars, the internode length is an important factor for machine harvest of tea leaves. More tea cultivars will be provided to draw firm conclusions. Therefore, in the future, more tea cultivar in different tea regions were provided to analyze the relationship and drawing the conclusion. Thank you for your valuable advice.

  1. The ratio of GAs and IAA also influences the internode length, so this parameter needs to be analyzed. 

Response: Thank you for your comments. The effect of the ratio of GAs and IAA on the internode length was analyzed in Discussion (Line 318-323).

  1. Discussion section needs to be re-written. 

Response: Thank you for your comments. As suggested, we have cited more literature and discussed it in the revised manuscript. In addition, logic has been re-organized. 

  1. 'variety' needs to be replaced by 'cultivar' all through the text.

Response: Thank you for your valuable suggestion. The 'variety' has been replaced by 'cultivar' through the text.

Reviewer 2 Report

In this study, 7 tea varieties were used to investi-12 gate the relationship between the internode length and blade angle of newly-formed tea shoots and 13 machine-harvest in the field experiments. Three stage field experiments showed that the internode length is an important factor for machine harvest of tea leaves, and the level of GA3 was strongly associated with internode length. The article is very interesting and current, however, it has some weaknesses.

1) Is the data in Figure 4 the mean? Is there any duplication? If so, it is better to display all duplicate samples.

2) Figure 5 and Figure 6 need to be adjusted, for the font and layout should be neat.

3) The scale in Figure 1 needs to be consistent.

4) The formulas in the text need to be labeled with numbers.

5) There are some minor errors in some units, such as kg ha-1.

6) The conclusion needs to be rewritten. The conclusion generally needs to include some overview, significance, and prospect of this study.

English language needs some improvements.

Author Response

Comments and Suggestions for Authors

In this study, 7 tea varieties were used to investigate the relationship between the internode length and blade angle of newly-formed tea shoots and machine-harvest in the field experiments. Three stage field experiments showed that the internode length is an important factor for machine harvest of tea leaves, and the level of GA3 was strongly associated with internode length. The article is very interesting and current, however, it has some weaknesses.

  • Is the data in Figure 4 the mean? Is there any duplication? If so, it is better to display all duplicate samples.

Response: Thank you for your suggestion. In figure 4, M and N indicate the standard rate of buds and leaves and the intact rate of buds and leaves, and the data in the heat map indicate the correlation coefficient between M or N and the internode length or blade angle of the seven tea cultivars.

  • Figure 5 and Figure 6 need to be adjusted, for the font and layout should be neat.

Response: Thank you for your suggestion. The Figure 5 and 6 have been adjusted to improve the uniformity.

  • The scale in Figure 1 needs to be consistent.
    Response: Thank you for your suggestion.In Figure 1, the scale in every variety have been regulated and the the length of ruler have been set 12 cm.
  • The formulas in the text need to be labeled with numbers.
    Response: Thank you for your suggestion.The formulas have been labeled (line 343-344).
  • There are some minor errors in some units, such as kg ha-1.
    Response: Thank you for your suggestion.The kg ha−1 has been revised kg ha-1.

  • The conclusion needs to be rewritten. The conclusion generally needs to include some overview, significance, and prospect of this study.
    Response: The conclusion has been rewritten.

Conclusions

Seven tea cultivars were used to investigate the relationship between the internode length or blade angles of newly formed tea shoots and the integrity of buds and leaves in field experiments conducted over one year. Longer internode length was strongly associated with the integrity of buds and leaves, and the level of GA3 was strongly related to internode length. Our study indicated that the internode length is crucial to the harvesting performance of tea-picking machines. Therefore, in future studies, tea cultivars with longer internodes will need to be cultivated to determine which cultivars are suitable for machine picking.

Reviewer 3 Report

Dear author,

After peer review I found the MS needs major revision and cannot be accepted in its present form. The MS may be revised in light of the comments below:

1.       The introduction needs to highlight the novelty of the study

2.       At the end part of the introduction research gap of the study is missing, author need to write little more elaborately and add some reports previously done relevant to the study.

3.       The paragraph is not required for introduction for line no. 79-83 in page no. 2. It may be shifted to relevant section.

4.       The discussion needs to be elaborated.

5.       The conclusion is not appropriate; a little description about the pros of the research work is needed. It should include major findings of the study. Redraft accordingly.

6.       The MS needs to be improved for English and editing in the framing the language is needed.

Dear author,

After peer review I found the MS needs to be improved for English and editing in the framing the language is needed.

Author Response

  1. The introduction needs to highlight the novelty of the study

Response: Thank you for your suggestion. The novelty of the study has been added at the end part of the introduction.

  1. At the end part of the introduction research gap of the study is missing, author need to write little more elaborately and add some reports previously done relevant to the study.

Response: Thank you for your suggestion. The novelty of the study and some reports previously done relevant to the study have been added into the revised manuscript at the end part of the introduction (line 79-92).

Zheng, X.X.; Ao, C.; Mao, X.X.; Cui, H.C.; Yu, J.Z. Preliminary study on selection on machine-picked varieties of Hangzhou high-quality tea. Zhejiang Agri. Sci. 2016,57 (5):661-663 (In Chinese).

Luo, Y.P.; Song, T.T.; Wen, D.H.;Tang, M.; Cai, W.Z. The internode length and blade angle of newly-formed tea shoots on machine picking. J. Zhejiang Univer.2009, 35(4): 420-424 (In Chinese).

  1. The paragraph is not required for introduction for line no. 79-83 in page no. 2. It may be shifted to relevant section.

Response: Thank you for your suggestion. The paragraph in line no. 79-83 have been deleted.

  1. The discussion needs to be elaborated.
    Response: Thank you for your comment. As suggested, we have cited more literature and discussed it in the revised manuscript.  

  1. The conclusion is not appropriate; a little description about the pros of the research work is needed. It should include major findings of the study. Redraft accordingly.

Response: Thank you for your suggestion. The conclusion has been rewritten, and the major findings of research work and prospect have been added into revised manuscript.

Conclusions

7 tea varieties were used to investigate the relationship between the internode length or blade angle of newly-formed tea shoots and the integrity of bud and leaves in the field experiments throughout the year. The longer internode length is strongly associated with the integrity of bud and leave, and the level of GA3 was strongly related with internode length. Our study indicated that the internode length is crucial on machine pick up performance of tea. Therefore, in future the tea varieties with longer internode were needed to be cultivated for suitable for machine picking.

  1. The MS needs to be improved for English and editing in the framing the language is needed.

Response: Thank you for your valuable suggestion. The manuscript has been polished by the MDPI to improve the presentation of this manuscript.

Round 2

Reviewer 1 Report

It can be accepted now.

Reviewer 3 Report

All the comments has been resolved properly now the MS is ready for publication 

Minor English  checking  required